# Physical Investigations on Bias-Free, Photo-Induced Hall Sensors Based on Pt/GaAs and Pt/Si Schottky Junctions

**DOI:** 10.3390/s21093009

**Published:** 2021-04-25

**Authors:** Xiaolei Wang, Xupeng Sun, Shuainan Cui, Qianqian Yang, Tianrui Zhai, Jinliang Zhao, Jinxiang Deng, Antonio Ruotolo

**Affiliations:** 1College of Physics and Optoelectronics, Faculty of Science, Beijing University of Technology, Beijing 100124, China; Sunxp@emails.bjut.edu.cn (X.S.); cuisn202006096@emails.bjut.edu.cn (S.C.); yangqianqian@bjut.edu.cn (Q.Y.); trzhai@bjut.edu.cn (T.Z.); zhaojinliang@bjut.edu.cn (J.Z.); jdeng@bjut.edu.cn (J.D.); 2Department of Natural Sciences, Florida Polytechnic University, 4700 Research Way, Lakeland, FL 33805, USA; aruotolo@floridapoly.edu

**Keywords:** Schottky junction, barrier height, photo-induced Hall effect, magnetic sensor

## Abstract

Hall-effect in semiconductors has wide applications for magnetic field sensing. Yet, a standard Hall sensor retains two problems: its linearity is affected by the non-uniformity of the current distribution; the sensitivity is bias-dependent, with linearity decreasing with increasing bias current. In order to improve the performance, we here propose a novel structure which realizes bias-free, photo-induced Hall sensors. The system consists of a semi-transparent metal Pt and a semiconductor Si or GaAs to form a Schottky contact. We systematically compared the photo-induced Schottky behaviors and Hall effects without net current flowing, depending on various magnetic fields, light intensities and wavelengths of Pt/GaAs and Pt/Si junctions. The electrical characteristics of the Schottky photo-diodes were fitted to obtain the barrier height as a function of light intensity. We show that the open-circuit Hall voltage of Pt/GaAs junction is orders of magnitude lower than that of Pt/Si, and the barrier height of GaAs is smaller. It should be attributed to the surface states in GaAs which block the carrier drifting. This work not only realizes the physical investigations of photo-induced Hall effects in Pt/GaAs and Pt/Si Schottky junctions, but also opens a new pathway for bias-free magnetic sensing with high linearity and sensitivity comparing to commercial Hall-sensors.

## 1. Introduction

The Hall effect in semiconductors has been used for more than one century to detect the intensity of magnetic fields [1]. Although magneto-resistive sensors in digital systems have been developed in recent years, semiconductor Hall sensors still retain irreplaceable in analog applications because of their two specific features: (1) they are intrinsically linear, and do not need magnetic materials; (2) their sensitivity is directly proportional to the bias current [2,3,4,5,6,7]. Yet, commercial Hall sensors have to operate by using a very low bias current, because linearity decreases with increasing bias-current [8]. On the contrary, the Hall effect also exists in metals. The low resistance of metals could increase uniformity of the current density and reduce dependence of linearity on bias current [9,10]. However, the transverse voltage is too small to have any practical application [11].

Lateral effect, Schottky photo-diodes are commonly used as magnetic- and position sensing- detectors [12]. Photo-excitation of carriers in a semiconductor can increase the Hall-voltage without increasing bias current. The photo-induced Hall effect in a Schottky diode, consisting of “metal/semiconductor” hetero-structure, includes the following physical processes: photo-generating in semiconductor, carriers separating and drifting across the interface, then diffusing in metal driven by a magnetic force [13]. A photon with energy larger than the band-gap of semiconductors could be absorbed during the photo-generation. It could transfer an electron from valence band to conduction band, or a hole from conduction band to valence band. Separation of carriers is achieved through the space charge region of a Schottky barrier, which relies on the built-in electric field. The carriers in semiconductor are injected into metal at a very high velocity, much faster than diffusion [13]. In this case, if a magnetic field is applied in the plane of a Schottky diode, it can generate large Lorentz forces and produce a transverse, open-circuit voltage at the metal edges that is proportional to the field, as well as light intensity. For a traditional lateral photo-diode, light is photo-converted into the increasing of an existed electrical current in a standard device [14,15,16,17]. Specifically, it consist of an extended p-type/intrinsic/n-type (p-i-n) semiconductor tri-layer with a bottom cathode electrode and four top lateral anodes. The p-type layer is thin enough to allow penetrating of light and resistive enough to avoid flowing of in-plane current through low doping concentration of acceptors. On the other side, the n-type layer is highly doped with donors in order to easily collect the current through the cathode. Therefore, the traditional lateral-effect photo-diode is a close-circuit, in which light beam causes a change of current flowing between the anodes and the cathode. However, current state-of-the-art, non-uniformity of the in-plane current distribution compromises linearity in these kinds of sensors. In contrast, in our device, light only reduces the built-in potential of the Schottky-barrier, with no net current flowing. This allows our sensor to operate in open-circuit conditions to recover linearity without increasing cost.

It has been recently reported [13,18,19,20] that a giant photo-Hall effect is induced in a novel device structure as mentioned above. A typical system consists of an ultra-thin Pt film that forms a Schottky contact to intrinsic Si. Compared to a standard Hall sensor, in-plane current is replaced by light, the magnetic field is applied in plane instead of perpendicular. It shows high linearity and sensitivity which are comparable to commercial Hall-sensors. Moreover, there is no net current in the circuit, so its performances are not affected by bias current.

## 2. Materials and Methods

It is necessary to engineer and optimize this novel Hall-sensor based on photo-induced Hall effect. Various Schottky bilayers could be explored to fabricate bias-free, optically-tunable Hall sensors [21]. The possibility to optimize the photo-Hall effect should be investigated by replacing the normal metal or the semiconductor. In order to form a highly rectifying Schottky contact, the metal is usually Au or Pt, which have work functions of ϕAu = 5.5 eV and ϕPt = 5.9 eV, respectively [22]. So Pt should be the best choice in this system. Instead, replacing semiconductor Si with a direct band-gap semiconductor, such as GaAs [23], may help to understand the physical mechanism and even increase sensitivity of the bias-free, photo-induced Hall sensors.

In order to verify the above idea, we carried out systematical experiments on Pt/GaAs and Pt/Si junctions. The intrinsic GaAs and Si wafers used in this work were single-side polished, (100)-oriented, with resistivity greater than 10,000 Ω·cm. The carrier concentration of Si is 1.5 × 10^10^ cm^−3^, while the carrier concentration of GaAs is 1.8 × 10^6^ cm^−3^. The semi-transparent Pt films of ~3 nm were grown by high vacuum sputtering from a 99.99% pure Pt target, with DC power supply 50 W. The base pressure was 2 × 10^−6^ Pa and the process gas (Ar) pressure was 0.2 Pa. The growth rate was calculated to be 0.15 nm/s and 20 s were used for our deposition. The ultra-thin Pt films were grown on both entire 2-inch GaAs and Si wafers, and then were cut into chips for current-voltage (I–V) characterizations and photo-Hall measurements. Before the top Pt layer deposition, small part of GaAs or Si surface was partially masked in order to access the bottom electrode of a Schottky diode. The size of Pt film in a Schottky junction was kept to be 1 × 1 cm^2^. Electrical contacts of the Schottky device were made by using silver paste to avoid the puncture of Pt thin film. The open-circuit Hall voltage was obtained by using a Keithley nanovoltmeter under an electromagnet.

## 3. Results

### 3.1. Schottky Rectifying Behaviors

Figure 1a shows the representation for our typical Schottky device of Pt/GaAs or Pt/Si junction in this work. Since the intrinsic wafer is highly resistive, the electrical setup could reveal the existence of a Schottky barrier at the interface. A significant series resistance (R_S_) is added by the GaAs or Si that can be subtracted to estimate the barrier height. Figure 2b shows that the light source is a commercial optical fiber illuminator emitting non-polarized, white light with a wavelength spectrum ranging from λ = 400 nm (violet) to λ = 800 nm (red), with maximum power 150 W. Light was uniformly shed on the sample.

Figure 1c,d show the physical origin of a Schottky photo-diode according to the theory of image forces [24]. If the metal has a work-function much higher than the semiconductor, a Schottky junction is formed at the interface [25]. Electrons move from the semiconductor to the metal, leaving uncompensated holes in the semiconductor. At equilibrium, the built-in potential V_bi_ prevents additional carriers to migrate. This equilibrium corresponds to the alignment of the Fermi levels, as shown in Figure 1c. When the sample is illuminated, light can penetrate through the semi-transparent metal and electron-hole pairs are generated [26]. Electrons can readily flow from the metal to the semiconductor to re-establish equilibrium, but electrons can neither overcome the barrier nor flow through the intrinsic semiconductor. Yet, an additional electron in the space charge region results in lowering of the barrier height, which can be modeled by the theory of image forces [27]. Considering an electron trapped at a distance z from the interface, a positive mirror image charge is induced in the metal at a distance –z. A second electric field builds up, with an opposite sign as compared to the built-in potential. The image potential energy associated with this field is given by the following Equation (1) and decreases with the distance z from the interface. It results in a rounding of the net potential barrier as depicted in Figure 1d.
(1)$$\phi \left(z\right)=-\int_{x}^{\infty }E_{I}\left(z\right)dz=\frac{q}{16\pi \varepsilon z}\;$$

Despite the significant series resistance offered by the GaAs or Si substrate, both I–V characteristics are clearly rectifying in Figure 2a,b. The current of Pt/GaAs junction is about one order of magnitude lower than that of Pt/Si, which may originate from the larger resistivity of GaAs. We show the influence of different light powers on Schottky behavior of both samples. Larger light intensity could excite more carriers in a semiconductor, so it increases the current of the Schottky junction and improve the rectifying behavior.

### 3.2. Photo-Induced Hall Effects

For typical Hall effect in metals, a given current is injected along the x-direction and magnetic field is applied along the z-direction. Hall voltage *V_H_* is measured along y-direction, as indicated in Figure 3a. *V_H_* is inversely proportional to the carrier density *n*. As *n* is very large in metals, *V_H_* is very small and difficult to be detected [28]. A novel photo-Hall sensor was devoted to increase the Hall effect in metals to magnitudes suitable for applications [29]. Figure 3b shows the schematic representation of photo-Hall effect by injecting light-excited carriers at high velocity through the interface of a metal/semiconductor Schottky contact.

The light source was the same one as Figure 1b and uniformly shed on the sample along the z-direction, shown as in Figure 3b, with the magnetic field applied along the x-direction and the open-circuit voltage measured along y-direction.

We engineered a similar system in Figure 3b, in which the Hall effect in Pt is greatly enhanced and dependence of linearity on bias current is suppressed. The device recovers tunable sensitivity, without compromising linearity, by replacing current bias with light bias. Light reaching the interface provides the force to drive carriers across the bilayer. A magnetic field applied in the junction plane exerts a magnetic force on the carriers diffusing in the metal, which accumulates on opposite sides, according to their sign. This carrier accumulation can be detected as a transverse, open-circuit voltage on the metal surface. *V_H_* is proportional to the field strength, as well as light intensity and wavelengths, as shown in Figure 4 and Figure 5. At equilibrium, the net current is zero regardless light intensity.

Photo-induced Hall measurements were carried out by applying light in different wavelengths ranges achieved by filters and full spectrum of the lamp, labeled as “no filter”, as magnetic field B = 0.3 T in Figure 4a,b. In the visible range, the photo-conversion increases from short wavelengths to longer ones. Different *V_H_* signals come from different photon-conversion energies. Noticeably, the open-circuit *V_H_* of Pt/GaAs shows orders of magnitude lower than *V_H_* of Pt/Si, which is out of expected. Similar results were obtained in magnetic field dependent of *V_H_*. As the field increases, so does the Lorentz force acting on the carriers and the photo-induced *V_H_*, indicated in Figure 4c,d.

In Figure 5, this effect was also found to be linear with both magnetic field strength and light intensity. We show the detected *V_H_* as a function of the field strength under different light intensities in Figure 5a,b. The voltage was proportional to the magnetic field strength. In Figure 5c,d, *V_H_* increased linearly with light intensity for fixed values of magnetic field. Following the same law in Figure 4, the effect depending on magnetic field and light intensity of Pt/GaAs shows orders of magnitude lower than *V_H_* of Pt/Si.

The comparison of the experimetal results on Pt/GaAs and Pt/Si permitted the deduction of surface state effects in GaAs, such as uncontrollable defects or capture centers [30]. Surface states are non-radiative in nature and can form traps in the energy structure, leading to a loss in the effective number of charge carriers penetrating the interface [31].

## 4. Discussion

In order to investigate the physical mechanism, we calculated the barrier height *φ_B_* as explained in the following. The I–V characteristic of a Schottky diode can be express as [32]:(2)$$I=I_{0}\left[\exp\left(\frac{e\left(V-R_{S}I\right)}{\eta k_{B}T}\right)-1\right]\;$$
where *R_S_* is a significant series resistance added by the semiconductor, *I*_0_ is the reverse bias saturation current, *e* is the electron charge, *η* is the ideality factor, *k_B_* is the Boltzmann constant, and *T* is the temperature.

Considering Equation (2) and Taylor’s series of the exponential function (e^x^ ≈ 1 + x) about the point *V* = 0, we can obtain:(3)$$I_{0}=\frac{\eta \frac{k_{B}T}{e}\frac{dI}{dV}}{1-R_{S}\frac{dI}{dV}}$$
where *R_S_* and *η* were estimated by deriving Equation (1) and considering the intercept and slope of the straight line which fits the curve:(4)$$\frac{dV}{d(\ln I)}=R_{S}I+\frac{\eta k_{B}T}{e}\;$$

The barrier height *ϕ_B_* can be estimated by using the following equation [33]:(5)$$\phi_{B}=\frac{k_{B}T}{e}\ln\left[\frac{AA^{*}T^{2}}{I_{0}}\right]$$
where *A* is the area of the device, *A** is the Richardson’s constant, that will be taken as *A** = 156 A cm^−2^ K^−2^ [34], and I_0_ is the saturation current in reverse bias.

The calculated values of barrier height *ϕ_B_* with respect to the light intensity for Pt/GaAs and Pt/Si Schottky junctions are reported in Figure 6. In our Hall sensor, *ϕ_B_* could determine the measured voltage, which is corresponding to the previous results.

From Figure 6, we found that the barrier height *ϕ_B_* of Pt/GaAs is lower than Pt/Si, and the change of *ϕ_B_* is much smaller for Pt/GaAs than for Pt/Si. It helps to demonstrate that GaAs has more surface trap states, which affect the performance of photo-induced Hall effect in our device. This issue could be alleviated by reducing the surface states through a passivation procedure in our future investigations.

Let us point out that the sensors presented here have a sensitivity of ~1 millVolt/(Watt Tesla), i.e., they would have the same sensitivity of a commercial Hall sensor if illuminated with 1 Watt, cold LED light. Besides, the sensitivity could be significantly increased by reducing the Pt thickness. Optimization of the sensors is beyond the scope of this work, in which we compare Si-based and GaAs-based sensors with same Pt thickness and light intensities.

## 5. Conclusions

In conclusion, we proposed photo-induced Hall effect in metal/semiconductor Schottky junctions as a new technique for magnetic field sensing and optical detection. Photo-excited carriers in a semiconductor drift fast to the metal, and are driven by Lorentz forces under magnetic field, then produce a transverse, open-circuit voltage at the metal edges without Joule heating. This photo-induced Hall voltage was proved to be proportional to the magnetic field, as well as light intensity and wavelength. The net current was zero, therefore linearity was not bias dependent. Comparison of Pt/GaAs and Pt/Si Schottky devices were carried out systematically by measuring photovoltaic I–V characteristics and bias-free, photo-induced Hall effects. The measurements of Pt/GaAs junction are all order of magnitude lower than that of Pt/Si, and the barrier height of GaAs is smaller after calculations. This work not only explores the physical origins of photo-induced Schottky and Hall effects, but also reveals the relationship between the surface trap states and device performance, which provides a new approach to bias-free magnetic and optical sensing application.

## Figures and Tables

**Figure 1 sensors-21-03009-f001:**
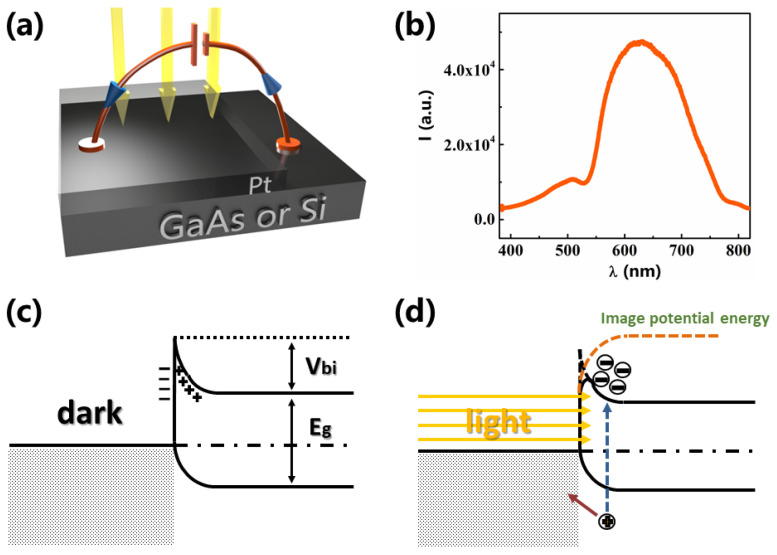
(**a**) The configuration for our typical Schottky device of Pt/GaAs or Pt/Si. (**b**) The spectrum of the light we used, covering all visible range. (**c**) Representation of the equilibrium corresponding to the alignment of the Fermi levels for a Schottky junction. V_bi_ is the theoretically calculated built-in potential. (**d**) Sketch of the photo-induced excitation of carriers and lowering of the barrier due to image forces.

**Figure 2 sensors-21-03009-f002:**
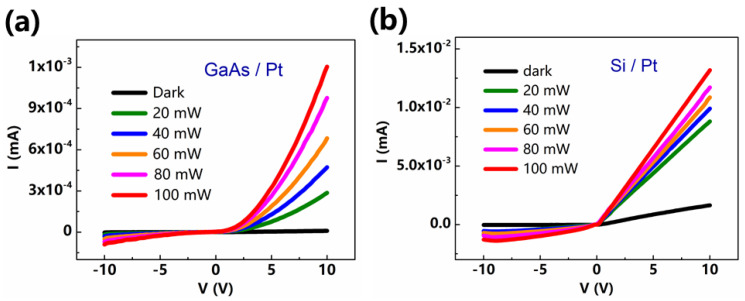
Current-voltage (I–V) characteristics of (**a**) Pt/GaAs and (**b**) Pt/Si bilayers in the dark and under different light intensities. Light intensities of 20 mW, 40 mW, 60 mW, 80 mW and 100 mW were detected on the sample surface.

**Figure 3 sensors-21-03009-f003:**
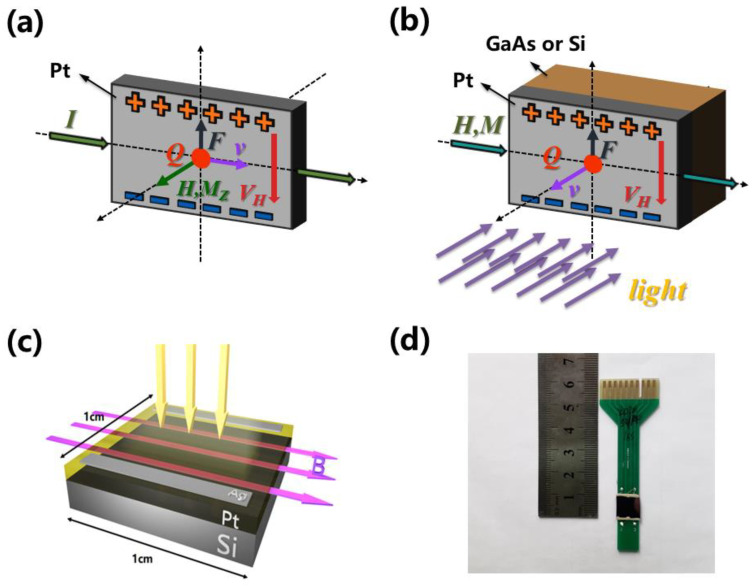
(**a**) Hall effect in a Pt film; (**b**) Photo-induced Hall effect in a Schottky photo-diode; (**c**) The configuration of our photo-Hall device under magnetic field; (**d**) The picture of our real device and sample holder for electrical connection.

**Figure 4 sensors-21-03009-f004:**
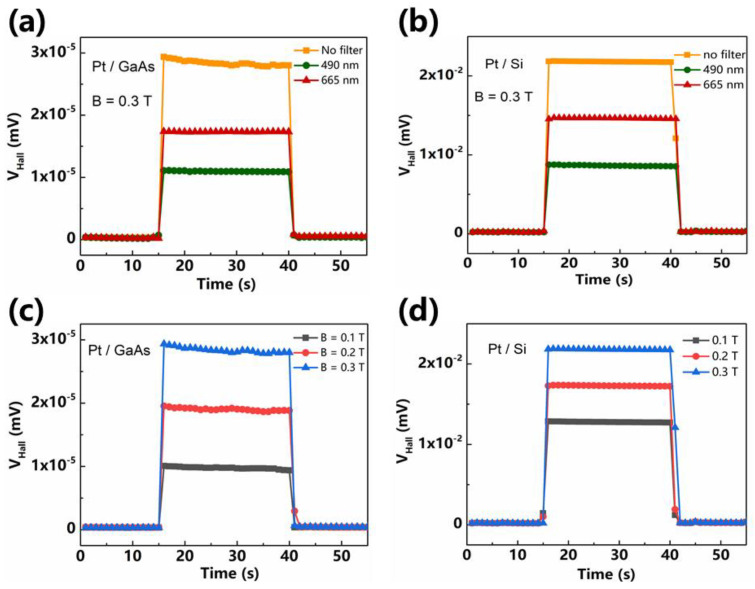
Photo-excitation of transverse, photo-induced Hall voltage vs. time elicited when exposing the (**a**) Pt/GaAs and (**b**) Pt/Si bilayers to light in different wavelength ranges as magnetic field B = 0.3 T. Similar signals were obtained under different magnetic fields B = 0.1, 0.2, and 0.3 T for (**c**) Pt/GaAs and (**d**) Pt/Si junctions under a constant while light. The power of light source in all the measurements was kept P = 50 mW.

**Figure 5 sensors-21-03009-f005:**
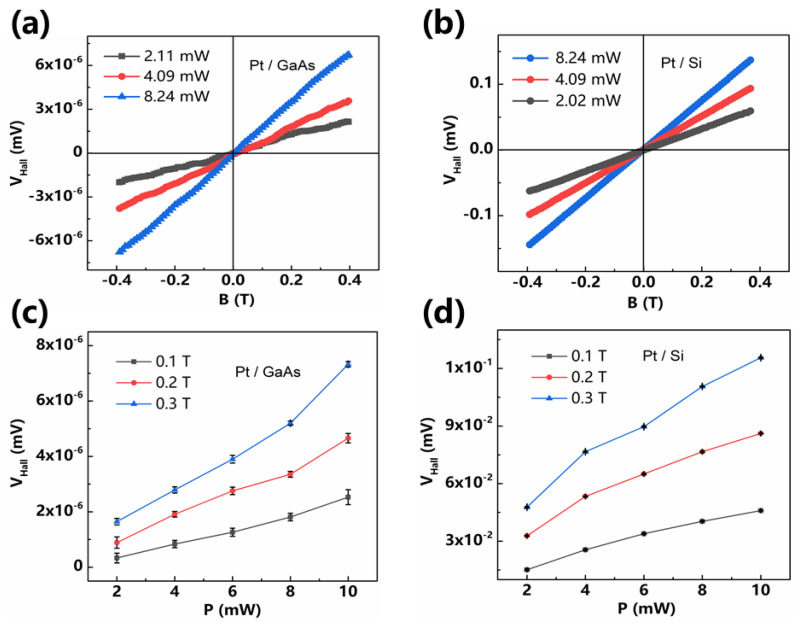
*V_H_*-*B* at constant light intensities for (**a**) Pt/GaAs and (**b**) Pt/Si. *V_H_*-*P* at constant magnetic fields of (**c**) Pt/GaAs and (**d**) Pt/Si with error bars.

**Figure 6 sensors-21-03009-f006:**
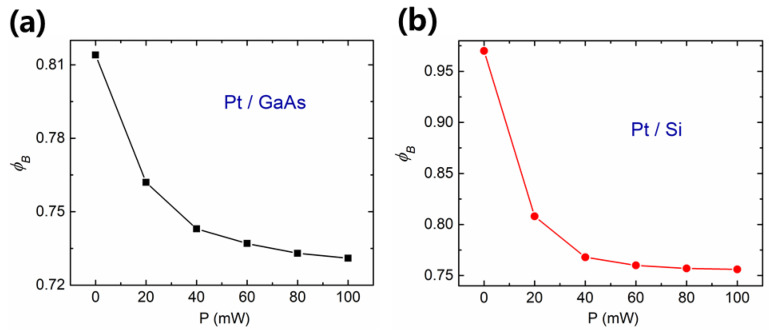
Barrier height as a function of the light intensity of (**a**) Pt/GaAs and (**b**) Pt/Si Schottky junctions.

## Data Availability

The data presented in this study are available in the article.

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
