# Peer review of "Physical Investigations on Bias-Free, Photo-Induced Hall Sensors Based on Pt/GaAs and Pt/Si Schottky Junctions"

_sensors, 2021, doi:10.3390/s21093009_

Round 1

Reviewer 1 Report

The paper present the physical properties of a photo-induced Hall effect in metal/semiconductor Schottky junctions.

The comparison with the state-of-the-art is poor and related more to the physics of the device itself than to the Hall sensors.  Same for itself paper and this is not good in my point of view for a Journal paper named "Sensor"
Notice that no limit of pages in MDPI publications. So please increase the references by >3X. This to properly comment on the state-of-the-art referring to previous/recent papers on the Hall effect published on Sensors, Electronic MDPI and IEEE on the basis of the practical use of the Hall effect sensor (practical application, pro, and contra in using them, etc.) 
This should be done with also to describe the effectiveness/improvement of the proposed sensor compared to the state-of-the-art and with the purpose to better differentiate the use and constraints of different Hall sensors during their respective use. In particular, this is needed in order to better define the practical use as a sensor of the Hall device presented in the paper (in terms of physics). Even from the present reference list is clear that the profile of the paper is more towards materials/physics instead of practical application as a sensor. As "Sensor" is the name of the journal, a more concrete engineering profile should be given to the paper. 

Please highlight also in the caption of each figure you are presenting measurement results. 

I personally do not see in the paper any effective improvement of the state-of-the-art and neither unexpected/innovative/remarkable results.

Author Response

Reply: We thank the referee for the comments. We believe the referee’s objection was coming from a misunderstanding of the scope of "Sensors". Physical Sensors is one of the main sections of this journal, covering all aspects of subjects for physical sensors, sensor technology and new physical sensor principles. Here we propose a novel sensor based on Hall effect, which can be photo-induced in metals and used for bias-free magnetic sensing. In order to prove the importance and advantage of our sensor, we stated the non-conventional mechanism in details, including photo-generation and injection of charge through a Schottky barrier. All the experimental measurements and physical investigations have demonstrated strongly that our system offers higher sensitivity and larger linearity range. For a traditional lateral photo-diode, light is photo-converted into an electrical current flowing in a standard device.  In contrast, light only reduces the built-in potential of the Schottky-barrier in our device, with no net current flowing. This allows the sensor to operate in open-circuit conditions to recover linearity without increasing cost. As the reviewer suggested, we have added the above descriptions about the innovation and improvement of the proposed sensor compared to the state-of-the-art.

Besides, we have increased the references by >3X according to the referee’s suggestion.

Reviewer 2 Report

Recommendation: Publish after minor revisions noted.

Comments: 

The manuscript is devoted to the detailed study of bias-free, photo-induced Hall sensors based on Pt/GaAs and Pt/Si Schottky junctions. The authors provided systematic and physical investigations on these two heterojunctions, which opens a new pathway for bias-free magnetic sensing. This work is well fulfilled and I recommend publishing it in Sensors. I have three recommendations as follows

1. How to determine the light power (or intensity) on samples in this work?

2. The authors claim that the linearity and sensitivity in their device is higher than commercial Hall-sensors, because of lower bias current and no Joule heating. Then, how are linearity and sensitivity influenced by light bias or magnetic field in their devices?

3. The authors attribute the smaller photo-induced Halleffect of Pt/GaAs to surface states on GaAs. What could be the surface states? The authors should provide more detailed information.

Author Response

We appreciate the referee’s positive evaluation of our manuscript.

  1. How to determine the light power (or intensity) on samples in this work?

Reply: The way we do it is to measure the light intensity with a power meter when put it at exactly the sample position, and multiply the obtained power density by sample area. 

  1. The authors claim thatthe linearity and sensitivity in their device is higher than commercial Hall-sensors, because of lower bias current and no Joule heating. Then, how are linearity and sensitivity influenced by light bias or magnetic field in their devices?

Reply: As shown in Figures 4 and 5, we have provided a linear relationship between Hall voltage and generated magnetic field at constant light intensities, while Hall voltage also increased linearly with light intensity for fixed values of magnetic field. So our devices show a large range of linearity, in addition to sensitivity to fluctuations of light bias or magnetic field.

  1. The authors attribute the smaller photo-induced Hall effect of Pt/GaAs to surface states on GaAs. What could be the surface states? The authors should provide more detailed information.

Reply: Normally, the GaAs surface is terminated with Ga or As atoms, and the surface state density is high due to its intrinsic defects and surface oxidation. So some passivation processes are needed to obtain a better surface with direct band gap. However, the exact surface states should be a bit complicated and it probably has mixed factors. So it is not accurate to draw a simple conclusion here. In our future works, we will be happy to do more systematic investigations to get sufficient evidences, in order to provide more detailed information about the surface states.

Reviewer 3 Report

The manuscript discusses a photo-induced Hall-effect magnetic sensor based on a Schottky barrier. The idea discussed is really interesting and could have an important impact on both the scientific community and industry. However, the idea was already presented by the authors in paper ([13] in the manuscript) and the novelty with respect to that paper is only on the comparison between Pt/Si barrier and Pt/GaAs barrier, from my understanding. I think that the novelty content is not sufficient for a full paper. I suggest the authors consider the publication of the work in a journal letter or at a conference.

 Apart from novelty content, I would like to open a scientific discussion on some technical points of the work:

  • The authors catastrophize the commercial Hall-effect sensors by saying their linearity is “largely affected by the non-uniformity of the current distribution” and their sensitivity is “severely compromised by Joule heating”. I agree with the authors on the presence of these two effects but they are not so critical. Linearity is usually better than 1% error and joule heating is usually neglected in commercial sensors. Can you better support your statement?
  • In section I, the authors state that Hall sensors are intrinsically linear (page 1 line 40-41). I agree with the authors but this statement is in contrast with what is said in the abstract.
  • The authors state that one advantage of their sensor is that it requires only 2 contacts. Actually, I think that you have to close the Schottky diode in some circuit in order to polarize it.
  • The authors state that no Joule heating is present but they do not provide any demonstration of this statement. In my opinion, the proposed sensor still shows the Joule heating effect though it is very low and, perhaps, negligible. Indeed, there is a current flowing through the Schottky barrier and this current is certainly associated with Joule heating. Given the reported sensitivities, I think that this current is very low and the heating is negligible. Can the authors demonstrate I’m wrong?
  • From a practical standpoint, the sensor requires a light source to work and its output is strongly dependent on it. Thus, on one hand, if the sensor is used in ambient light without a dedicated light source then the sensor will suffer from a strong cross-sensitivity. On the other hand, if a dedicated light source with constant power is used, then this affects the cost and implementation of the system. I would like that the authors comment on this.
  • The sensitivity that I can roughly estimate from Fig. 4 is 10 uV/T, while a standard Hall sensor could have more than 100 mV/T for 1 mA bias (see [1]). This could be problematic in real scenario implementations. Can the authors comment on this?  
  • How was created the magnetic field used to get the static characteristics?
  • Figures need more details:
    • Is the voltage reported in the abscissa of fig 2 the polarization voltage of the diode? This must be stated clearly.
    • What voltage bias was used for figure 4
    • What wavelength was used for figures 4-c and 4-d?
    • Contrary to what stated in the manuscript, Figures 5-c and 5-d report a non-linear relationship, in my opinion. Please explain. I would like to know how much is the non-linearity of these curves.
  • Is the ideality factor missing in eq. 3 or did you set it equal to 1?
  • Please explain the difference between I_0 and I_0*

[1]          B. Liu et al., “Low-power and high-sensitivity system-on-chip hall effect sensor,” in 2017 IEEE SENSORS, 2017, pp. 1–3.

Reviewer 4 Report

The authors present physical investigations on bias-free, photo-induced Hall sensors based on Pt/GaAs and Pt/Si Schottky junctions. The authors directly deposited Pt contacts on Si and GaAs wafers to fabricate hall-induces hall sensor. I have serious concerns with the novelty of this paper which needs to be justified by the authors in the manuscript and in the response letter. Hence, I would like to decide its fate based on the major revision of the comments in the next. 

1- Remove introduction first paragraph. This also establishes the fact that the paper need serious revisions for typos and text errors.

2- Provide the novelty of this study and the most important research findings in the last introduction section. Also, to what application did the authors fabricated these photodetectors?

3- The authors have simply used the commercially available GaAs and Si wafers and deposited the metal electrodes in the laboratory. How would the authors justify the novelty of this device with commercial wafers and simple metal contacts?

4- What is the analogy of depositing Ag paste (a metal) over an already deposited Pt layer? Don't you think that it would only increase the contact resistance and does nothing good to the device?

5- Figure 2: Why there is a very large current shift increasing the photo power from dark (Zero) to 20 mW compared to the rest of the intensities that witnessed a small shift in current intensity?

6- Figure 2(b) shows an ohmic behavior to the positive voltage bias, which is against the very concept of this paper. Why there is no resistance to the built-in barrier in this case, which is always expected of a Schottky contact? 

Round 2

Reviewer 1 Report

Ok. the authors' reply convinced me.s

Author Response

 We thank the referee for the acceptance of our manuscript to be published in Sensors.

Reviewer 3 Report

  • The introduction is now more balanced. Thank you for accepting my suggestion. However, I still doubt about the prominence of Joule heating with nonlinearity effects. In my opinion, the heating of the Hall probe due to the bias current is negligible. A typical silicon Hall sensor has a resistance of roughly 1 kOhm, thus it will dissipate around 1 mW when biased with 1 mA. Given the modern package technology, this power consumption gives origin to a thermal variation of a few K (I considered a junction-to-ambient thermal resistance of 100 K/W, which is quite standard for plastic SOIC package). Therefore, I don’t think the joule heating has a direct effect on linearity of the Hall sensors.

However, I also did a literature research to prove my doubt. I started from the two articles cited by the authors, i.e. [7] and [8], but they do not discuss Joule heating in Hall sensors, so I’m also asking the authors why they cited these two articles. Then, I made a quick search and I found only a recent paper by Dowling et. al. [R1] discussing the effect of Joule heating on the temperature dispersion of the offset in GaN Hall sensors. This article somehow confirmed my theory, reporting local temperature variation of 6 degree Celsius, maximum. Finally, I checked commercial devices starting from the one cited by the authors. In this case, I can see that the Joule heating could be more prominent, and I agree with the authors that the company reports the nonlinearity error only at the nominal bias and temperature. I must also cite the sensors from Allegro, which are intended for current sensing and integrate a current-carrying trace on the sensor chip. In this case, the current to be sensed can be as high as 20 A and it will certainly originate an important Joule heating. However, the sensors nonlinearity is rated below 2 % throughout the full input range. Therefore, state-of-the-art Hall sensors have solved Joule heating problem.

In summary, I suggest the authors to better describe the problem, the current state of the art, and better positioning their novelty.

  • No further comments.

  • Thank you for better explaining the behavior of your device but I still have a question: Hall effect works on moving carriers, thus you should have a kind of current somewhere. As far as my understanding, the light reduces the built-in potential of the Schottky barrier, allowing the electrons activated by the photons to be injected into the semiconductor at very high speed. These electrons should be related to a kind of heating effect. I kindly ask the authors to better describe this point.

  • I still would like to see a measurement demonstrating the absence of Joule heating.

  • Thank you for your response. Could you kindly add a measurement result showing that the ambient light has no effects?

  • Thank you for your response. I would just ask you to double check your numbers for sensitivity. From fig. 4b (yellow line) I can roughly estimate a peak value of 2E-2 mV for an input mag field of 0.3 T, giving a sensitivity of 66 uV/T. Please check numbers.

[R1] K. M. Dowling et al., “Low Offset and Noise in High Biased GaN 2DEG Hall-Effect Plates Investigated with Infrared Microscopy,” J. Microelectromechanical Syst., vol. 29, no. 5, pp. 669–676, Oct. 2020.

Author Response

We are grateful to the reviewer for this useful discussion. We have tried and checked the sensitivity of the Allegro sensor mentioned by the reviewer but we find a sensitivity in V/A, rather than V/T. One can always keep Joule heating low if the current-carrying wire is far away from the Hall sensor. Of course, 20A will still generate enough field to be sensed. We are probably straying too far in this useful discussion.

We have replaced "loss of linearity because of Joule heating" with "linearity decreasing with increasing bias current" (which is a fact, as reported in data sheets) without specifying the complex reasons of this dependency (which must involve, amongst other effects, Joule heating if heat sinks are suggested). For example, in the abstract we write: "the sensitivity is bias-dependent, with linearity decreasing with increasing bias current.” We have also rephrased introduction and conclusions. Changes are highlighted in the manuscript. Joule heating is never specifically mentioned. This should be a safer claim while waiting for further investigations on the complex reasons (non-uniformity of current, Joule heating, etc.) of the loss of linearity with increasing bias current in traditional Hall sensors.

Thanks again for the careful review and constructive suggestions to improve our manuscript. We have replaced the references [6-8] and added more descriptions to  better position the novelty of our manuscript.

Reviewer 4 Report

The authors have responded well to all the queries raised in review round 1. I want the authors to add the provided explanations in the response letter into the main body of the manuscript as well for readers' understanding.

Good Luck

Author Response

We appreciate the referee’s high evaluation. We have added the provided explanations in the response letter into the main body of the manuscript now.

Round 3

Reviewer 3 Report

The authors have not answered to all my concerns.

Please look at my previous report. 

Author Response

Please see the attachment, which could show our replies more clearly. We have made point to point reply in the respond letter this time. 
